# The Journey of Action Recognition

Xi Ding
Australian National University
Canberra, Australian Capital Territory, Australia
Xi.Ding1@anu.edu.au

Lei Wang*
Griffith University
Brisbane, Queensland, Australia
Australian National University
Canberra, Australian Capital Territory, Australia
l.wang4@griffith.edu.au

## Abstract

Action recognition has evolved from a niche research area into a fundamental aspect of video understanding, driven by the dynamic interplay between data, model architectures, and learning paradigms. Early studies, constrained by limited datasets and handcrafted features, laid the groundwork for the field, but the rapid growth of data and advancements in deep learning techniques ignited a revolution. From 2D- and 3D-CNNs to spatiotemporal graph convolutional networks, these models have advanced the ability to capture complex, multidimensional actions across increasingly diverse and multimodal datasets. Simultaneously, innovative learning paradigms such as self-supervised, few-shot, and zero-shot learning have transformed how we use data, enabling models to generalize across tasks with minimal labeled data. The advent of transformer-based architectures has catalyzed a new era in action recognition, excelling in capturing long-range temporal dependencies and overcoming previous limitations in spatiotemporal modeling. Furthermore, the rise of video masked autoencoders has introduced new ways to balance spatial and temporal information, leading to breakthroughs in understanding motion dynamics. This paper presents a comprehensive exploration of action recognition through three critical lenses: the evolution of model architectures, the expanding diversity of data, and the emergence of innovative learning techniques. By tracing the trajectory of these developments, we highlight how the convergence of these elements has broadened the scope of action recognition to tackle more complex video processing challenges, including anomaly detection, captioning, and video question answering. In particular, we underscore the transformative role of large language models in infusing semantic context, significantly enhancing the performance and versatility of action recognition systems. Our work not only reflects on the past but also provides a roadmap for future advancements. We reveal how action recognition has transcended its original focus, positioning itself at the heart of general video analysis. By synthesizing these insights, we offer a forward-thinking perspective on how the integration of multimodal, temporal, and semantic information will shape the future of AI-powered video understanding. Our paper's GitHub repository can be found here.

*Corresponding author.

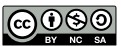

*WWW Companion '25, April 28-May 2, 2025, Sydney, NSW, Australia*
© 2025 Copyright held by the owner/author(s).
ACM ISBN 979-8-4007-1331-6/2025/04
https://doi.org/10.1145/3701716.3717746

## CCS Concepts

• **Computing methodologies** → *Computer vision representations*; **Learning paradigms**; **Machine learning algorithms**; **Activity recognition and understanding**; • **Networks** → **Network architectures**.

## Keywords

Action recognition, Data, Model architectures, Learning paradigm

**ACM Reference Format:**
Xi Ding and Lei Wang. 2025. The Journey of Action Recognition. In *Companion Proceedings of the ACM Web Conference 2025 (WWW Companion '25), April 28-May 2, 2025, Sydney, NSW, Australia.* ACM, New York, NY, USA, 16 pages. https://doi.org/10.1145/3701716.3717746

## 1 Introduction

Action recognition, the task of identifying and understanding human actions in video, has become a pivotal area of research in computer vision and machine learning [85]. It plays a crucial role in a wide range of applications, from surveillance systems and autonomous driving to video indexing and human-computer interaction. Early research works in action recognition focus on small, labeled video datasets and rely heavily on handcrafted features to capture motion and spatial information [9, 48, 117, 199, 202, 250]. However, with the rapid growth of video data and advancements in machine learning techniques, the field has undergone significant transformations, leading to more robust, scalable, and accurate methods [13, 21, 234, 250, 265].

The evolution of action recognition can be understood through three key interconnected dimensions: the data, the learning paradigms, and the model architectures. As datasets grow in scale and complexity, researchers begin to shift from simple, labeled datasets to more diverse and larger video repositories. This shift enables the development of learned representations through deep learning techniques, which outperform traditional methods that rely on handcrafted features [32, 49, 101, 223, 312]. Concurrently, new learning paradigms, such as unsupervised, self-supervised, and few-shot learning, are introduced to better use the expanding volume of unlabeled video data [41, 50, 76, 81]. These paradigms enable models to generalize more effectively, making it possible to learn action recognition tasks without the need for large amounts of manually labeled data.

In parallel, model architectures evolve from simple 2D convolutional networks (CNNs) to more complex 3D and two-stream networks designed to capture spatial and temporal features [33, 183, 215, 231]. Recent advancements in transformer-based models and video masked autoencoders have further pushed the boundaries of action recognition, allowing for better handling of long-range

temporal dependencies and improving the capture of both spatial and temporal motion features [70, 230, 251, 278]. The integration of language models and vision-language models into action recognition has further enhanced the field, enabling richer contextual understanding of actions and their relationships to textual descriptions [106, 161, 273].

This paper aims to provide a comprehensive exploration of the journey of action recognition from its early stages to its current state. We discuss the evolution of the field from both data and model perspectives and examine how learning paradigms have shaped the progress of action recognition research. Through this analysis, we uncover valuable insights into the challenges, breakthroughs, and future directions of action recognition, as it continues to advance and become an integral part of broader video processing tasks. The main **contributions** of this paper are as follows:

  i. A detailed review of the evolution of action recognition from data, learning, and model perspectives, highlighting key milestones and breakthroughs.
 ii. An in-depth exploration of the co-evolution of paradigms, data, and architectures, offering a unified view of the interdependencies that have shaped the field.
iii. A discussion on the future directions and emerging trends in action recognition, emphasizing the integration of multimodal data, transformer-based architectures, and vision-language models, and their potential to address the challenges in video understanding and processing.

## 2 Related Work

Action recognition has garnered substantial attention over the past few decades, resulting in numerous survey papers that examine the evolution of methods, datasets, and models [53, 109, 225, 249, 252]. These surveys provide valuable insights into the historical progression of action recognition, categorize various approaches, and identify the challenges and future directions. However, despite the wealth of surveys, each focuses on different aspects of the problem, and most either emphasize specific models, learning paradigms, or datasets, without providing a comprehensive analysis of the interplay between data, learning methods, and models.

Early surveys on action recognition primarily focus on the progress of handcrafted features. These papers, including [252] and [225], provide a thorough examination of early video descriptors and their effectiveness for action recognition in the context of small labeled datasets. They explore how different feature extraction methods contributed to the performance of action recognition models and discuss the limitations of traditional methods when applied to large-scale datasets. While informative, these surveys do not emphasize the evolving role of deep learning, and therefore miss the pivotal transition from handcrafted features to learned representations that would shape future advancements in the field.

As deep learning began to dominate, surveys such as [111] and [80] explore the impact of convolutional neural networks (CNNs), 3D CNNs, and two-stream architectures on action recognition. These works review the evolution from simple 2D architectures to more complex 3D and temporal models, which incorporate the crucial temporal dimension alongside spatial features. They also focus on datasets like UCF101, HMDB51, and Kinetics, which play a

significant role in advancing the field. While these surveys provide in-depth analyses of different architectures, they are often limited in scope, concentrating mainly on model innovations without exploring the full spectrum of learning paradigms, such as unsupervised and self-supervised learning, that would later play a crucial role in overcoming the challenges posed by limited labeled data.

In more recent years, there has been an increasing interest in action recognition models that use large-scale multimodal datasets and emerging learning paradigms, including self-supervised learning, few-shot learning, and transfer learning. Surveys such as [175], [167], and [277] provide insights into these newer approaches and highlight how they use large video datasets, *e.g.*, Kinetics-400, to improve action recognition performance. These works focus on the advantages of training models on large, diverse datasets and discuss the trade-offs between supervised and unsupervised learning. However, they often treat learning paradigms and model architectures as separate topics, without sufficiently considering how they co-evolve in tandem with the increasing complexity of data. A few recent surveys, such as [46, 95, 100, 115], have started to address the role of transformers and vision-language models in action recognition, emphasizing the growing importance of incorporating semantic context into video understanding. These works explore how transformers, with their ability to model long-range dependencies, have become a powerful tool for capturing temporal dynamics in action recognition tasks. While these surveys acknowledge the synergy between models like BERT or GPT and vision models, they generally do not delve deeply into how these models interact with the data and learning paradigms to shape the development of action recognition.

**Differences from existing work.** While existing surveys on action recognition have made significant contributions by examining various aspects of the field, there are key differences in the scope and focus of this work. First, this paper takes a more holistic approach by integrating the perspectives of data, learning paradigms, and model architectures in a unified framework. Rather than treating each component in isolation, we explore how they co-evolve and influence one another, offering a comprehensive understanding of the factors driving advancements in action recognition.

Second, this paper provides a deeper exploration of the evolution of learning paradigms in action recognition, including the transition from supervised learning to unsupervised, self-supervised, and few-shot learning. This is an important distinction, as it highlights the increasing reliance on large-scale unlabeled datasets and the emergence of pretraining techniques, which are essential for handling the complexities of modern video datasets. Unlike many existing surveys that focus primarily on model architectures or specific datasets, this work emphasizes the shifting learning paradigms that are enabling the field to scale and generalize.

Finally, this work incorporates a forward-looking perspective by discussing the integration of vision-language models and transformers in action recognition, which are still underexplored in existing surveys. While other surveys mention these advances, they often fail to address their potential for cross-modal learning and the broader impact on video processing tasks. This paper not only examines the impact of transformers and language models on

**Table 1: The journey of action recognition (Part 1): Methods based on RGB videos, including handcrafted features, 2D CNNs, (2+1)D CNNs, 3D CNNs, two-stream networks, and transformers. Columns detail learning paradigms, data modalities, and publication venues (year).**

| | Method | Venue | Learning | Dataset | Modality |
|---|---|---|---|---|---|
| **Handcrafted** | HL-STIP[117] | IJCV 2005 | Supervised | Outdoor scenes [117] | RGB |
| | Spatio-temporal Cuboids[48] | VS-PETS 2005 | Supervised | Human Action Dataset[201] | RGB |
| | 3D-SURF[202] | ECCV 2006 | Supervised | Mikolajczyk[163] | RGB |
| | NNMF Detector [284] | ICCV 2007 | Supervised | KTH[201] | RGB |
| | HOG3D[107] | BMVC 2008 | Supervised | KTH[201], Weizmann[74], Hollywood[118] | RGB |
| | Laptev et al.[118] | CVPR 2008 | Supervised | KTH[201] | RGB+Optical flow |
| | Action MACH[192] | CVPR 2008 | Supervised | KTH[201], Weizmann[74] | RGB |
| | Extended SURF[282] | ECCV 2008 | Supervised | KTH[201], TRECVID 2006[179] | RGB |
| | LTP[309] | ICCV 2009 | Supervised | UCF101[221] | RGB |
| | Messing et al.[159] | ICCV 2009 | Supervised | KTH[201], Hollywood[118], Kissing and slapping dataset[192], UCF Sports[192] | RGB |
| | Bregonzio et al.[16] | CVPR 2009 | Supervised | KTH[201], Weizmann[74] | RGB |
| | Tracklet Descriptors [190] | ECCV 2010 | Supervised | KTH[201], ADL[159], Hollywood[118] | RGB+Optical flow |
| | Dense Long-Duration Trajectories[224] | ICME 2010 | Supervised | KTH[201] | RGB+Optical flow |
| | Dense Trajectories[240] | IJCV 2013 | Supervised | KTH[201], YouTube[141], Hollywood2[155], UCF Sports[192], IXMAS[280], Olympic Sports[171], UCF50[191], UIUC[232], HMDB51[114] | RGB+Optical flow |
| | iDT[242] | ICCV 2013 | Supervised | Hollywood2[155], HMDB51[114], Olympic Sports[171], UCF50[191] | RGB+Optical flow |
| | Taylor videos [266] | ICML 2024 | Supervised | HMDB51[114],CATER[71],MPII Cooking[193], Kinetics-400[103], -600[19], Something-Something V2[77],NTU RGB+D[142, 205], Kinetics-skeleton[295] | RGB+Skeleton |
| **2D-based** | Slow fusion[101] | CVPR 2014 | Supervised | Sports-1M[101], UCF101[221] | RGB |
| | CNN-LSTM[312] | CVPR 2015 | Supervised | Sports-1M[101], UCF101[221] | RGB+Optical flow |
| | LRCN[49] | CVPR 2015 | Supervised | UCF101[221] | RGB+Optical flow |
| | Composite LSTM[223] | ICML 2016 | Unsupervised | UCF101[221], HMDB51[114] | RGB |
| | Rank Pooling[63] | TPAMI 2016 | Supervised | HMDB51[114], Hollywood2[155], MPII Cooking[193] | RGB+Optical flow |
| | LENN[67] | CVPR 2016 | Supervised | UCF101[221] | RGB |
| | Bilen et al.[14] | TPAMI 2017 | Supervised | UCF101[221], HMDB51[114] | RGB |
| | TSN[265] | TPAMI 2018 | Supervised | HMDB51[114], UCF101[221], Kinetics-400[103], ActivityNet[18], THUMOS14[91] | RGB+RGB differences+Optical flow+Warped optical flow+Audio |
| | Attention-LSTM[152] | CVPR 2018 | Supervised | UCF101[221], HMDB51[114], Kinetics-400[103] | RGB+Optical flow+Audio |
| | PEAR[288] | ICME 2019 | Reinforcement | UCF101[221], Sports-1M[101] | RGB+Optical flow |
| | TSM[134] | ICCV 2019 | Supervised | Something-Something V1[77], Something-Something V2[77], Kinetics-400[103], UCF101[221], HMDB51[114] | RGB |
| | VINCE[73] | arXiv 2020 | Self-supervised | Kinetics-400[103] | RGB |
| | C²LSTM[154] | Neurocomputing 2020 | Supervised | UCF101[221], HMDB51[114] | RGB |
| | MoCo[61] | CVPR 2021 | Self-supervised | Kinetics-400[103], UCF101[221], HMDB51[114] | RGB |
| | TCL[216] | CVPR 2021 | Semi-supervised+Contrastive | Mini-Something-V2[23], Kinetics-400[103], Charades-Ego[213] | RGB |
| | TDN[263] | CVPR 2021 | Supervised | Something-Something V1[77], Something-Something V2[77], Kinetics-400[103] | RGB |
| | DB-LSTM[83] | Neurocomputing 2021 | Supervised | UCF101[221], HMDB51[114] | RGB+Optical flow |
| | SeCo[308] | AAAI 2021 | Self-supervised | Kinetics-400[103], UCF101[221], HMDB51[114], ActivityNet[18] | RGB |
| | Xiao et al.[286] | CVPR 2022 | Semi-supervised+Contrastive | Kinetics-400[103], UCF101[221], HMDB51[114], Kinetics-400[103] | RGB |
| | GCSM[310] | ACM MM 2023 | Few-shot | UCF101[221], HMDB51[114], Kinetics-400[103] | RGB |
| | GgHM[290] | ICCV 2023 | Few-shot | HMDB51[114], UCF101[221], Kinetics-400[103], Something-Something V2[77] | RGB |
| **3D-based** | C3D[231] | ICCV 2015 | Supervised | UCF101[221] | RGB |
| | I3D[21] | CVPR 2017 | Supervised | Kinetics-400[103], UCF101[221], HMDB51[114] | RGB |
| | P3D[183] | ICCV 2017 | Supervised | Sports-1M[101], UCF101[221], ActivityNet[18] | RGB |
| | ResNet3D[82] | CVPR 2018 | Supervised | Kinetics-400[103], UCF101[221], HMDB51[114], ActivityNet[18] | RGB |
| | S3D[289] | ECCV 2018 | Supervised | Kinetics-400[103], Something-Something V1[77], UCF101[221], HMDB51[114] | RGB+Optical flow |
| | CSN[233] | ICCV 2019 | Supervised | Sports-1M[101], Kinetics-400[103], Something-Something V1[77] | RGB |
| | SlowFast[60] | ICCV 2019 | Supervised | Kinetics-400[103], Something-Something V1[77], Kinetics-600[19], Charades[214], AVA[79] | RGB |
| | STM[94] | ICCV 2019 | Supervised | Something-Something V1[77], Something-Something V2[77], Kinetics-400[103], UCF101[221], HMDB51[114] | RGB |
| | DEEP-HAL [258] | CVPR 2019 | Self-supervised | HMDB51[114], Charades[214],MPII Cooking[193] | RGB |
| | Xv et al.[292] | CVPR 2019 | Supervised | UCF101[221], HMDB51[114] | RGB+Optical flow |
| | X3D[58] | CVPR 2020 | Supervised | Kinetics-400[103], Kinetics-600[19], Charades[214], AVA[79] | RGB |
| | TPN[298] | CVPR 2020 | Supervised | Kinetics-400[103], Something-Something V1[77], Something-Something V2[77], Epic-Kitchens[38] | RGB |
| | SpeedNet[12] | CVPR 2020 | Self-supervised | Kinetics-400[103], UCF101[221], HMDB51[114], NfS[66] | RGB |
| | CoCLR[81] | NeurIPS 2020 | Self-supervised | UCF101[221], HMDB51[114], Kinetics-400[103] | RGB+Optical flow |
| | VTHCL[297] | arXiv 2020 | Self-supervised | Kinetics-400[103], UCF101[221], HMDB51[114] | RGB+Optical flow |
| | MvPL[291] | ICCV 2021 | Semi-supervised | Kinetics-400[103], UCF101[221], HMDB51[114] | RGB |
| | CVRL[181] | CVPR 2021 | Self-supervised | Kinetics-400[103], Kinetics-600[19], UCF101[221], HMDB51[114] | RGB |
| | Yang et al.[302] | CVPR 2021 | Supervised | Kinetics-400[103], Kinetics-700[20], Charades[214], Something-Something V1[77], AVA[79] | RGB |
| | 3DResNet+ATFR[57] | CVPR 2021 | Supervised | Kinetics-400[103], Kinetics-600[19], UCF101[221], HMDB51[114], Something-Something V2[77] | RGB |
| | MoViNet[108] | CVPR 2021 | Supervised | Kinetics-400[103], Kinetics-600[19], Kinetics-700[20], Something-Something V2[77], Epic-Kitchens-100[39], MiT[165], Charades[214] | RGB |
| | ODF+SDF [253] | ACM MM 2021 | Reinforcement+Zero-shot | HMDB51[114], Charades[214],MPII Cooking[193], EPIC-Kitchen[38] | RGB+Optical flow + object / saliency detectors |
| | CLASTER [76] | ECCV 2022 | Zero-shot | UCF101[221], HMDB51[114], Olympic Sports[171] | RGB+Optical flow+Semantic embeddings |
| | TFCNet[316] | arXiv 2022 | Supervised | Diving48[133], CATER[71] | RGB |
| | Multi-Transforms[237] | ICMEW 2024 | Self-supervised | UCF101[221], HMDB51[114] | RGB |
| | HoT [262] | ICASSP 2024 | Supervised | HMDB51[114],MPII Cooking[193] | RGB+Optical flow |
| | Flow corr. [257] | ICASSP 2024 | Supervised | HMDB51[114], Charades[214],MPII Cooking[193] | RGB+Optical flow |
| **Two-stream** | Two-Stream ConvNet[215] | NeurIPS 2014 | Supervised | UCF101[221], HMDB51[114] | RGB+Optical flow |
| | P-CNN[33] | ICCV 2015 | Supervised | JHMDB[93], MPII Cooking[193] | RGB+Optical Flow+Joint |
| | TDD[261] | CVPR 2015 | Supervised | HMDB51[114], UCF101[221] | RGB+Optical flow |
| | Two-Stream Fusion[62] | CVPR 2016 | Supervised | UCF101[221], HMDB51[114] | RGB+RGB differences+Optical flow+Warped optical flow |
| | TSN-Two-Stream[264] | ECCV 2016 | Supervised | HMDB51[114], UCF101[221] | RGB+Optical flow |
| | DOVF[116] | CVPR 2017 | Supervised | UCF101[221], HMDB51[114] | RGB+Optical flow |
| | TLE[44] | CVPR 2017 | Supervised | UCF101[221], HMDB51[114] | RGB+Optical flow |
| | ActionVLAD[72] | CVPR 2017 | Supervised | HMDB51[114], UCF101[221], Charades[214] | RGB+Optical flow |
| | TRN-Two-Stream[321] | ECCV 2018 | Supervised | Something-Something V1[77], Something-Something V2[77], Charades[214] | RGB |
| | TSM-Two-Stream[134] | ICCV 2019 | Supervised | Something-Something V1[77], Something-Something V2[77], Kinetics-400[103], UCF101[221], HMDB51[114] | RGB |
| | KTSN[146] | arXiv 2020 | Supervised | FSD-10 [46] | RGB+Optical flow+Skeleton |
| | MSM-ResNets[326] | IVC 2021 | Supervised | UCF101[221], HMDB51[114] | RGB+Optical Flow+Motion Saliency |
| | MAT-EffNet[320] | MMSys 2023 | Supervised | UCF101[221], HMDB51[114], Kinetics-400[103] | RGB+Optical flow |
| | TTFA[41] | SPL 2024 | Few-shot | Something-Something V2[77], Kinetics-400[103] | RGB |
| **(2+1)D-based** | R(2+1)D[234] | CVPR 2018 | Supervised | Kinetics-400[103], Sports-1M[101], UCF101[221], HMDB51[114] | RGB+Optical flow |
| | R(2+1)D+BERT[99] | ECCVW 2020 | Supervised | HMDB51[114], UCF101[221] | RGB |
| | XDC[6] | NeurIPS 2020 | Self-supervised | Kinetics-400[103], UCF101[221], HMDB51[114] | RGB+Optical flow+Audio |
| | ELo[180] | CVPR 2020 | Self-supervised | Kinetics-400[103], UCF101[221], HMDB51[114] | RGB |
| | Jin et al.[97] | ICICSP 2021 | Supervised | UCF101[221] | RGB+Audio |
| | GDT[176] | arXiv 2021 | Supervised | Kinetics-400[103], UCF101[221], HMDB51[114] | RGB+Audio |
| | AVID[166] | CVPR 2021 | Supervised | Kinetics-400[103], UCF101[221], HMDB51[114] | RGB+Audio |
| **Transformer-based** | VTN[170] | ICCV 2021 | Supervised | Kinetics-400[103], MiT[165] | RGB |
| | TimeSformer[13] | ICML 2021 | Supervised | Kinetics-400[103], Kinetics-600[19] | RGB |
| | STAM[207] | arXiv 2021 | Supervised | Kinetics-400[103], UCF101[221], Charades[214] | RGB |
| | ViViT[7] | ICCV 2021 | Supervised | Kinetics-400[103], Kinetics-600[19], Epic-Kitchens-100[39], MiT[165], Something-Something V2[77] | RGB |
| | MViT[54] | ICCV 2021 | Supervised | Kinetics-400[103], Kinetics-600[19], Something-Something V2[77], Charades[214], AVA[79] | RGB |
| | Motionformer[177] | NeurIPS 2021 | Supervised | Kinetics-400[103], Something-Something V2[77], Epic-Kitchens-100[39] | RGB |
| | X-ViT[17] | NeurIPS 2021 | Supervised | Kinetics-400[103], Kinetics-600[19], Something-Something V2[77], Epic-Kitchens-100[39] | RGB |
| | TallFormer[29] | ECCV 2022 | Supervised | THUMOS14[91], ActivityNet[18] | RGB |
| | VideoSwin[150] | CVPR 2022 | Supervised | Kinetics-400[103], Kinetics-600[19], Something-Something V2[77] | RGB |
| | ORViT[86] | CVPR 2022 | Supervised | Something-Something V2[77], SomethingElse[156], Diving48[133], AVA[79], Epic-Kitchens-100[39] | RGB |
| | BEVT[270] | CVPR 2022 | Self-supervised | Kinetics-400[103], Something-Something V2[77], Diving-48[133] | RGB |
| | MaskFeat[278] | CVPR 2022 | Self-supervised | Kinetics-400[103], Kinetics-600[19], Kinetics-700[20] | RGB |
| | Uniformer[125] | arXiv 2022 | Supervised | Kinetics-400[103], Kinetics-600[19], Something-Something V1[77], Something-Something V2[77] | RGB |
| | VideoMAE[230] | NeurIPS 2022 | Self-supervised | Something-Something V2[77], UCF101[221], HMDB51[114], AVA[79] | RGB |
| | MTV[294] | CVPR 2022 | Supervised | Kinetics-400[103], Kinetics-600[19], Kinetics-700[20], Something-Something V2[77], Epic-Kitchens-100[39], MiT[165] | RGB |
| | MAE-ST[59] | arXiv 2022 | Self-supervised | Kinetics-400[103], Something-Something V2[77], AVA[79] | RGB |
| | CAST[120] | NeurIPS 2023 | Supervised | Kinetics-400[103], Something-Something V2[77], Epic-Kitchens-100[39] | RGB |
| | UniFormerV2[126] | ICCV 2023 | Supervised+Contrastive | Kinetics-400[103], Kinetics-600[19], Kinetics-700[20], MiT[165], Something-Something V1[77], Something-Something V2[77], ActivityNet[18], HACS[319] | RGB |
| | OmniMAE[70] | CVPR 2023 | Self-supervised | Something-Something V2[77], Epic-Kitchens-100[39], Kinetics-400[103] | RGB |
| | MVD[271] | CVPR 2023 | Self-supervised | Kinetics-400[103], Something-Something V2[77], UCF101[221], HMDB51[114] | RGB |
| | Hiera[196] | ICML 2023 | Self-supervised | Kinetics-400[103], Kinetics-600[19], Kinetics-700[20], Something-Something V2[77], AVA[79] | RGB |
| | VideoMAE V2[251] | CVPR 2023 | Self-supervised | Something-Something V2[77], Kinetics-400[103], UCF101[221], HMDB51[114] | RGB |
| | SOAP[88] | ACM MM 2024 | Few-shot | Something-Something V2[77], Kinetics-400[103], UCF101[221], HMDB51[114] | RGB |
| | C2C[129] | ECCV 2024 | Zero-shot | Sth-com[129] | RGB |
| | VMPs [25] | ACML 2024 | Self-supervised | HMDB51[114],MPII Cooking 2[194], FineGym [206] | RGB+Motion prompts |
| | TIME Layer [24] | arXiv 2024 | Self-supervised | UCF101[221], HMDB51[114], UWA3D Multiview Activity II[186], NTU RGB+D[205], NTU RGB+D 120[142] | RGB+Depth |

action recognition, but also explores how these developments contribute to the broader landscape of video understanding, anomaly detection, captioning, and beyond.

## 3 From Actions to Insights

In this section, we systematically explore the evolution of action recognition through the interconnected lenses of data, model architectures, and learning paradigms. We delve into how each perspective has driven advancements in the field, while highlighting their co-evolution, showcasing how innovations in one domain have influenced and been shaped by progress in the others. Tables 1 and 2 show the progression of action recognition methods, while Table 3 highlights the evolution of action recognition datasets.

### 3.1 From a Data Perspective

The development of action recognition has been fundamentally shaped by the evolution of datasets, which act as the foundation for learning paradigms and model architectures. This journey shows a dynamic interplay between the characteristics of data and technological advancements in extracting meaningful patterns, leading to a continuous refinement of methods.

**Data evolution and paradigm shifts.** Early datasets like KTH [201], Hollywood2 [155], and Olympic Sports [171] mark the initial phase of action recognition research. These datasets, collected in controlled environments, feature a limited number of subjects and simple actions such as walking, waving, or running. Their simplicity inspires researchers to focus on handcrafted features [159, 190, 242],

**Table 2: The journey of action recognition (Part 2): Methods using alternative modalities, including skeleton-based, depth-based, infrared-based, point cloud-based, and multi-modal approaches (*e.g.*, text or audio). Columns detail learning paradigms, data modalities, and publication venues (year).**

| | Method | Venue | Learning | Dataset | Modality |
|---|---|---|---|---|---|
| | Dynamic Skeletons [87] | CVPR 2015 | Supervised | MSRDailyActivity[247], CAD-60[227], SYSU 3D HOI[87] | Depth+Joint |
| | HBRNN-L [52] | CVPR 2015 | Supervised | MSRAction3D[132], Berkeley MHAD[173], HDM05[168] | Joint |
| | Part-aware LSTM[205] | CVPR 2016 | Supervised | NTU RGB+D[205] | RGB+Depth+Joint+Infrared |
| | LARP-SO[236] | CVPR 2016 | Supervised | Florence3D-Action[203], MSRActionPairs3D[174], G3D-Gaming[15] | Joint |
| | STA-LSTM [218] | AAAI 2017 | Supervised | NTU RGB+D[205] | Joint |
| | LieNet [90] | CVPR 2017 | Supervised | NTU RGB+D[205], HDM05[168], G3D-Gaming[15] | Joint+Bone |
| | Two-Stream RNN [243] | CVPR 2017 | Supervised | NTU RGB+D[205] | Joint |
| | Ke et al. [104] | CVPR 2017 | Supervised | NTU RGB+D[205] | Joint |
| | VA-LSTM [315] | ICCV 2017 | Supervised | NTU RGB+D[205], SYSU 3D HOI[87] | Joint |
| | View Invariant[145] | Pattern Recognit. 2017 | Supervised | NTU RGB+D[205], Northwestern-UCLA[248], UWA3D Multiview Activity II[186], MSRC-12[64] | Joint |
| | Two-Stream CNN[123] | ICMEW 2017 | Supervised | NTU RGB+D[205], PKU-MMD I[137] | Joint+Skeleton motion |
| | LSTM-CNN[122] | ICMEW 2017 | Supervised | NTU RGB+D[205] | Joint |
| | ST-LSTM+Trust Gate [143] | TPAMI 2017 | Supervised | NTU RGB+D[205], MSRAction3D[132], SYSU 3D HOI[87], Berkeley MHAD[173] | Joint |
| | ST-GCN[295] | AAAI 2018 | Supervised | Kinetics-400[103], NTU RGB+D[205] | Joint |
| | Tang et al. [229] | CVPR 2018 | Reinforcement | NTU RGB+D[205], SYSU 3D HOI[87], UTKinect-Action3D[285] | Joint+Bone |
| | AS-GCN [128] | CVPR 2019 | Supervised | NTU RGB+D[205], Kinetics-400[103] | Joint+Bone |
| | 2s-AGCN[211] | CVPR 2019 | Fully-supervised | Kinetics-skeleton[295] | Joint+Bone |
| | DGNN [210] | CVPR 2019 | Supervised | Kinetics-skeleton[295] | Joint+Bone |
| | EfficientGCN[219] | ACM MM 2020 | Supervised | NTU RGB+D[205], NTU RGB+D 120[142] | Joint+Velocity+Bone |
| | RA-GCN [220] | TCSVT 2020 | Supervised | NTU RGB+D[205], NTU RGB+D 120[142] | RA-GCN |
| | Shift-GCN [30] | CVPR 2020 | Supervised | NTU RGB+D[205], NTU RGB+D 120[142], Northwestern-UCLA[248] | Joint+Bone |
| | MS-G3D [151] | CVPR 2020 | Supervised | NTU RGB+D 60[205], NTU RGB+D 120[142], Kinetics-skeleton[295] | Joint+Bone |
| Skeleton-based | DSTA-Net [212] | ACCV 2020 | Supervised | NTU RGB+D[205], NTU RGB+D 120[142] | Joint+Bone |
| | SCK+DCK / SCK⊕+DCK⊕ [110] | TPAMI 2020 | Supervised | UTKinect-Action3D[285], Florence3D-Action[203],MSRAction3D[132],NTU RGB+D 60[205], Kinetics-400[103], HMDB51[114], MPII Cooking[193] | Joint |
| | CTR-GCN[27] | ICCV 2021 | Supervised | NTU RGB+D 120[142], Northwestern-UCLA[248] | Joint+Bone |
| | FGCN [301] | TIP 2022 | Supervised | NTU RGB+D[205], NTU RGB+D120[142], Northwestern-UCLA[248] | Joint+Bone |
| | AGE-Ens [182] | TNNLS 2022 | Supervised | NTU RGB+D[205], NTU RGB+D 120[142] | Joint+Bone |
| | PoseConv3D[53] | CVPR 2022 | Supervised | Kinetics-400[103], UCF101[221], HMDB51[114] | Joint+Bone+RGB |
| | InfoGCN [34] | CVPR 2022 | Supervised | NTU RGB+D[205], NTU RGB+D 120[142], Northwestern-UCLA[248] | Joint+Bone |
| | DASTM[153] | ECCV 2022 | Few-shot | NTU RGB+D 120[142], Kinetics-skeleton[295] | Joint+Bone |
| | Uncertainty-DTW [255] | ECCV 2022 | Supervised/Unsupervised few-shot | NTU RGB+D[205], NTU RGB+D 120[142], Kinetics-skeleton[295] | Skeleton sequences |
| | TranSkeleton [139] | TCSVT 2023 | Supervised | NTU RGB+D[205], NTU RGB+D 120[142] | Joint+Bone |
| | HiCo [50] | AAAI 2023 | Unsupervised+Contrastive | NTU RGB+D[205], NTU RGB+D 120[142], PKU-MMD I[144], PKU-MMD II[144] | Joint |
| | FR-Head [322] | CVPR 2023 | Supervised+Contrastive | NTU RGB+D[205], NTU RGB+D 120[142], Northwestern-UCLA[248] | Joint+Bone |
| | 3Mformer [256] | CVPR 2023 | Supervised | NTU RGB+D[205], NTU RGB+D 120[142], Kinetics-400[103], Northwestern-UCLA[248] | Joint + Hyper-edge |
| | HYSP [148] | ICLR 2023 | Self-supervised | NTU RGB+D[205], NTU RGB+D 120[142], PKU-MMD I[144] | Joint |
| | PAINet[148] | ICCV 2023 | Few-shot | NTU RGB+D 120[142], Kinetics-skeleton[295] | Joint |
| | PCM³ [314] | ACM MM 2023 | Self-supervised | NTU RGB+D[205], NTU RGB+D 120[142], PKU-MMD I[144] | Joint+Bone+Motion |
| | Stream-GCN [304] | IJCAI 2023 | Supervised | NTU RGB+D[205], NTU RGB+D 120[142], Northwestern-UCLA[248] | Joint+Bone |
| | SkeletonGCL [89] | arXiv 2023 | Self-supervised | NTU RGB+D[205], NTU RGB+D 120[142], Northwestern-UCLA[248] | Joint+Bone |
| | DSCNet [31] | ESWA 2024 | Supervised+Multimodal | NTU RGB+D[205], NTU RGB+D 120[142], PKU-MMD I[144], UAV-Human[127], IKEA ASM[11], Northwestern-UCLA[248] | RGB+Joint+Bone |
| | Skeleton-OOD [293] | Neurocomputing 2024 | Supervised | NTU RGB+D[205], NTU RGB+D 120[142], Kinetics-400[103] | Joint |
| | ViA [300] | IJCV 2024 | Self-supervised | Posetics[299], NTU RGB+D[205], NTU RGB+D 120[142], Toyota Smarthome[40], UAV-Human[127], Penn Action[317] | Joint+Motion |
| | DeGCN [169] | TIP 2024 | Supervised | NTU RGB+D[205], NTU RGB+D 120[142], Northwestern-UCLA[248] | Joint+Bone+Motion |
| | Js-taPR-GCN[121] | TCSVT 2024 | Supervised | NTU RGB+D[205], NTU RGB+D 120[142], Northwestern-UCLA[248] | Joint+Bone+Motion |
| | BlockGCN [323] | CVPR 2024 | Supervised | NTU RGB+D[205], NTU RGB+D 120[142], Northwestern-UCLA[248] | Joint+Bone+Motion |
| | JEANIE [260] | IJCV 2024 | Supervised/Unsupervised few-shot | NTU RGB+D[205], NTU RGB+D 120[142], Kinetics-skeleton[295], MSRAction3D[132], UWA3D Multiview Activity[187] | Skeleton sequences |
| | SA-DVAE[130] | arXiv 2024 | Zero-shot | NTU RGB+D[205], NTU RGB+D 120[142], PKU-MMD[144] | Joint |
| | ProtoGCN [140] | arXiv 2024 | Self-supervised+Prototype | NTU RGB+D[205], NTU RGB+D 120[142], Kinetics-skeleton[295], FineGYM[206] | Joint |
| | HSIC-based[303] | arXiv 2024 | Supervised | NTU RGB+D[205], NTU RGB+D 120[142], Northwestern-UCLA[248] | Joint+Bone |
| | USDRL[281] | AAAI 2025 | Self-supervised | NTU RGB+D[205], NTU RGB+D 120[142], PKU-MMD I[144], PKU-MMD II[144] | Joint+Bone+Motion |
| | HON4D[174] | CVPR 2013 | Supervised | MSRAction3D[132], MSRDailyActivity3D[246], MSRActionPairs3D[174] | Depth |
| | HOPC[187] | ECCV 2014 | Supervised | MSRAction3D[132], MSRActionPairs3D[174], UWA3D Multiview Activity[187] | Depth+point cloud |
| | Wang et al.[268] | Trans. Human-Mach. Syst. 2016 | Supervised | MSRAction3D[132], MSRDailyActivity3D[246], UTKinect-Action3D[285] | Depth |
| | Rahmani et al.[188] | CVPR 2016 | Supervised | Northwestern-UCLA[248], UWA3D Multiview Activity II[186] | Depth |
| Depth-based | S²DDI[269] | ICCVW 2017 | Supervised | MSRAction3D[132], G3D-Gaming[15], MSRDailyActivity3D[246], SYSU 3D HOI[87], UTD-MHAD[22] | Depth |
| | Wang et al.[267] | TMM 2018 | Supervised | NTU RGB+D[205] | Depth |
| | MVDI[287] | Inf. Sci. 2018 | Supervised | NTU RGB+D[205], Northwestern-UCLA[248], UWA3D Multiview Activity II[186] | Depth |
| | 3DFCNN[197] | Multimed. Tools Appl. 2020 | Supervised | NTU RGB+D[205], Northwestern-UCLA[248], UWA3D Multiview Activity II[186] | Depth |
| | Liu et al.[138] | ICASSP 2017 | Supervised | MSRAction3D[132], DHA[136] | RGB+Depth |
| | Dhiman et al.[43] | TIP 2020 | Supervised | NTU RGB+D[205], UWA3D Multiview Activity II[186], Northwestern-UCLA[248] | Depth |
| | Stateful ConvLSTM[198] | arXiv 2020 | Supervised | NTU RGB+D[205] | Depth |
| | DEAR[189] | arXiv 2024 | Supervised | Something-Something V2[77] | RGB+Depth |
| | Gao et al.[68] | Neurocomputing 2016 | Supervised | InfAR[68] | Infrared+Optical flow |
| | Jiang et al.[96] | CVPRW 2017 | Supervised | InfAR[68] | Infrared+Optical flow |
| | Kawashima et al.[102] | AVSS 2017 | Supervised | Custom Dataset[102] | Infrared |
| | Shah et al.[204] | SPIE 2018 | Supervised | Custom IR Dataset[204] | Infrared |
| Infrared-based | TSTDDs[149] | SPL 2018 | Supervised | InfAR[68], NTU RGB+D[205] | Infrared+Optical flow |
| | Akula et al.[3] | CSR 2018 | Supervised | Custom IR Dataset[3] | Infrared |
| | Imran et al.[92] | Infrared Phys. Technol. 2019 | Supervised | InfAR[68], IITR-IAR[92] | Infrared+Optical flow |
| | Meglouli et al.[157] | CEAI 2019 | Supervised | InfAR[68] | Infrared+Optical flow |
| | Mehta et al.[158] | ICPR 2020 | Adversarial | TSF[235] | Infrared+Optical flow |
| | MeteorNet[147] | ICCV 2019 | Supervised | MSRAction3D[132] | Point cloud |
| | PointLSTM[164] | CVPR 2020 | Supervised | MSRAction3D[132] | Point cloud |
| | 3DV-PointNet++[275] | CVPR 2020 | Supervised | NTU RGB+D[205], NTU RGB+D 120[142], Northwestern-UCLA[248], UWA3D Multiview Activity II[186] | Depth |
| | ASTA3DConv[239] | Trans. Instrum. Meas. 2020 | Supervised | MSRAction3D[132] | Point cloud |
| Point cloud | Wang et al.[244] | WACV 2021 | Self-supervised | NTU RGB+D[205], NTU-PCL[205], MSRAction3D[132] | Point cloud |
| | P4Transformer[55] | CVPR 2021 | Supervised | MSRAction3D[132], NTU RGB+D[205], NTU RGB+D 120[142] | Point cloud |
| | PSTNet[56] | ICLR 2021 | Supervised | MSRAction3D[132], NTU RGB+D[205], NTU RGB+D 120[142] | Point cloud |
| | PST²[279] | WACV 2022 | Supervised | MSRAction3D[132] | Point cloud |
| | MaST-Pre[208] | ICCV 2023 | Self-supervised | MSRAction3D[132], NTU RGB+D[205] | Point cloud |
| | PointCPSC[209] | ICCV 2023 | Self-supervised | MSRAction3D[132], NTU RGB+D[205] | Point cloud |
| | 3DInAction[10] | CVPR 2024 | Supervised | MSRAction3D[132] | Point cloud |
| | KAN-HyperpointNet[28] | arXiv 2024 | Supervised | NTU RGB+D[205], MSRAction3D[132] | Point cloud |
| | CPD[131] | arXiv 2020 | Self-supervised | Kinetics-400[103], HMDB51[114], UCF101[221] | RGB+Text |
| | G-Blend[272] | CVPR 2020 | Multi-task | Kinetics-400[103], Mini-Sports[101], EPIC-Kitchen[38] | RGB+Optical flow+Audio |
| | MIL-NCE [161] | CVPR 2020 | Self-supervised | HowTo100M[162], HMDB51[114], UCF101[221] | RGB+Text |
| | MMV[5] | NeurIPS 2020 | Self-supervised | UCF101[221], HMDB51[114], Kinetics-600[19] | RGB+Audio+Text |
| | VIMPAC[228] | arXiv 2021 | Self-supervised | Something-Something V2[77], Diving48[133], UCF101[221], HMDB51[114] | RGB+Text |
| Text / Audio | InternVideo[274] | CVPR 2023 | Self-supervised | Kinetics-400[103], Kinetics-600[19], Kinetics-700[20], Something-Something V1[77], Something-Something V2[77], ActivityNet[18], HACS[319], HMDB51[114] | RGB+Text |
| | Side4Video[306] | arXiv 2023 | Self-supervised | Something-Something V1[77], Something-Something V2[77], Kinetics-400[103] | RGB+Text |
| | EZ-CLIP[2] | arXiv 2024 | Zero-shot | Kinetics-400[103], HMDB51[114], UCF101[221], Something-Something V2[77] | RGB+Text |
| | SATA[135] | arXiv 2024 | Zero-shot | UCF101[221], HMDB51[114] | RGB+Text |
| | TC-CLIP[106] | ECCV 2024 | Zero-shot/Few-shot/Fully-supervised | HMDB51[114], UCF101[221], Kinetics-400[103], Something-Something V2[77] | RGB+Text |
| | InternVideo2[273] | arXiv 2024 | Self-supervised+Multimodal | Kinetics-400[103], Kinetics-600[19], Kinetics-700[20], MiT[165], Something-Something V2[77], ActivityNet[18], HACS[319], Charades[214], HMDB51[114] | RGB+Audio+Text |
| | OmniVID[245] | CVPR 2024 | Supervised | Kinetics-400[103], Something-Something V2[77], UCF101[221], HMDB51[114] | RGB+Text |
| | LoCATe-GAT[200] | TETCI 2024 | Zero-shot | UCF101[221], HMDB51[114], ActivityNet[18], Kinetics-400[103] | RGB+Text |
| | STDD[311] | arXiv 2024 | Zero-shot | Kinetics-600[19], UCF101[221], HMDB51[114] | RGB+Text |

such as Histogram of Oriented Gradients (HOG) [37] and dense trajectories [241]. These manually crafted descriptors, combined with traditional classifiers like Support Vector Machines (SVMs), excel in recognizing these straightforward actions.

The next wave of datasets, such as HMDB51 [114], UCF101 [221], and Sports-1M [101], introduce more diversity in terms of actions, scenes, and contexts. The increased scale and variety requires a paradigm shift towards data-driven methods [152, 154, 223]. These datasets facilitate the adoption of deep learning, as convolutional neural networks (CNNs) could now exploit the broader representation power of larger and more complex datasets [305].

Larger-scale datasets like the Kinetics family [19, 20, 103], Something-Something V1 and V2[77], and Moments in Time [165] further push the field towards supervised learning. These datasets, with millions of labeled videos, provide the necessary foundation for deep models

to achieve state-of-the-art results [58, 108, 298]. However, the high cost of annotating video data leads to innovations in unsupervised and self-supervised learning. For instance, unlabeled datasets like HowTo100M [162] spur progress in contrastive learning approaches [61, 73, 81], while multimodal datasets, such as video-text pairs from ActivityNet Captions [113] and WebVid [8], enable breakthroughs in vision-language models like CLIP [184] and Flamingo [4]. These advancements demonstrate how the evolution of datasets directly influences paradigm shifts, from supervised learning to unsupervised, self-supervised, and multimodal approaches. Each paradigm addresses the growing complexity and scale of modern video data.

**Learning paradigms driven by data.** The nature of datasets plays a pivotal role in determining the choice of learning paradigms. Supervised learning thrives on large, labeled datasets, where explicit annotations like action labels provide clear supervision signals.

**Table 3: The journey of action recognition datasets: An overview of their evolution over time. This table includes detailed statistics, covering key aspects such as sensors, modalities, and characteristics, providing insights into their diversity and scope.**

| Datasets | Year | #Classes | #Subjects | #Views | #Video clips | Sensor | Modalities | Dataset type |
|---|---|---|---|---|---|---|---|---|
| KTH[201] | 2004 | 6 | 25 | 1 | 2391 | Static camera | RGB | Human actions (*e.g.*, walking, jogging) |
| Weizmann[74] | 2005 | 10 | 9 | 1 | 90 | - | RGB | Human actions (*e.g.*, jumping, running) |
| IXMAS[280] | 2006 | 11 | 10 | 5 | 330 | - | RGB | Movie Scenes (*e.g.*, kissing, running) |
| Hollywood[118] | 2008 | 8 | - | - | 1422 | - | RGB | Movie Scenes (*e.g.*, eating, driving) |
| Hollywood2[155] | 2009 | 12 | - | - | 1709 | - | RGB | Movie Scenes (*e.g.*, running, kissing) |
| ADL[160] | 2009 | 10 | 5 | - | 150 | Static camera | RGB | Daily Activities (*e.g.*, brushing teeth, reading) |
| Olympic Sports[171] | 2010 | 16 | - | - | 783 | - | RGB | Sports (*e.g.*, high jumping, diving) |
| MSRAction3D[132] | 2010 | 20 | 10 | 1 | 567 | Kinect v1 | Depth+3DJoints | Daily Activities (*e.g.*, drinking, walking) |
| CAD-60[227] | 2011 | 14 | 4 | - | 68 | Kinect v1 | RGB+Depth+3DJoints | Human performing activities (*e.g.*, cleaning objects) |
| HMDB51[114] | 2011 | 51 | - | - | 6,766 | - | RGB | Human actions (*e.g.*, jumping, running) |
| MSRDailyActivity3D[246] | 2012 | 16 | 10 | 1 | 320 | Kinect v1 | RGB+Depth+3DJoints | Daily Activities (*e.g.*, calling, playing game) |
| UCF101[221] | 2012 | 101 | - | - | 13,320 | - | RGB | Body motion, Human-object interactions, sports *etc.* |
| UTKinect-Action3D[285] | 2012 | 10 | 10 | 1 | 199 | Kinect v1 | RGB+Depth+3DJoints | Human actions (*e.g.*, waving hands, pushing) |
| MPII Cooking[193] | 2012 | 64 | 12 | 1 | 3,748 | - | RGB | Cooking |
| G3D-Gaming[15] | 2012 | 20 | 10 | 1 | - | Kinect v1 | RGB+Depth+3DJoints | Gaming scenario (*e.g.*, defending, climbing) |
| Berkeley MHAD[173] | 2013 | 11 | 12 | 4 | 660 | Multi-baseline stereo cameras | RGB+Depth+3DJoints+Accelerometer+Audio | Human actions (*e.g.*, throwing, clapping hands) |
| CAD-120[112] | 2013 | 10 | 4 | - | 120 | Kinect v1 | RGB+Depth+3DJoints | Human performing activities (*e.g.*, picking objects) |
| UCF50[191] | 2013 | 50 | - | - | 6676 | - | RGB | Body motion, Human-object interactions, sports *etc.* |
| Florence3D-Action[203] | 2013 | 9 | 10 | 1 | 215 | Kinect v1 | RGB+Depth+3DJoints | Human actions (*e.g.*, bowing, drinking) |
| MSRActionPairs3D[174] | 2013 | 12 | 10 | 1 | 360 | Kinect v1 | RGB+Depth+3DJoints | Human actions (*e.g.*, picking up, putting down) |
| Sports-1M[101] | 2014 | 487 | - | - | 1,000,000 | - | RGB | Sports (*e.g.*, swimming, skiing) |
| THUMOS14[91] | 2014 | 101 | - | - | 5,613 | - | RGB | Human Actions (*e.g.*, making up, archery) |
| Northwestern-UCLA[248] | 2014 | 10 | 10 | 3 | 1494 | Kinect v1 | RGB+Depth+3DJoints | Human actions (*e.g.*, dropping trash) |
| UWA3D Multiview Activity[187] | 2014 | 30 | 10 | 1 | 701 | Kinect v1 | RGB+Depth+3DJoints | Daily Activities (*e.g.*, holding head, walking) |
| ActivityNet[18] | 2015 | 203 | - | - | 27,801 | - | RGB | Human actions (*e.g.*, drawing, washing) |
| MPII Cooking 2[194] | 2015 | 67 | 30 | 1 | 273 | Static camera | RGB | Cooking |
| UWA3D Multiview Activity II[186] | 2015 | 30 | 9 | 4 | 1,070 | Kinect v1 | RGB+Depth+3DJoints Daily Activities | (*e.g.*, waving head, jumping) |
| SYSU 3D HOI[87] | 2015 | 12 | 40 | - | 480 | Kinect v1 | RGB+Depth+3DJoints | Human-Object Interactions (*e.g.*, sweeping the floor) |
| NTU RGB+D[205] | 2016 | 60 | 40 | 80 | 56,880 | Kinect v2 | RGB+Depth+3DJoints | Daily actions, health-realted actions *etc.* |
| InfAR[68] | 2016 | 12 | 40 | - | 600 | Infrared camera | Infrared | Human actions (*e.g.*, jogging) |
| TSF[235] | 2016 | 2 | - | 1 | 44 | FLIR ONE | Infrared | Falls and normal activities |
| Charades[214] | 2016 | 157 | - | - | 66,500 | - | RGB+Flow | Indoor activities (*e.g.*, cleaning) |
| PKU-MMD I[137] | 2017 | 51 | 66 | 3 | 1,076 | Kinect v2 | RGB+Depth+Infrared+3DJoints | Human actions (*e.g.*, walking) |
| NfS[66] | 2017 | - | - | - | 100 | 240 FPS camera | RGB | Visual object tracking |
| Kinetics-400[103] | 2017 | 400 | - | - | 306,245 | - | RGB | Human-centered actions (*e.g.*, playing instruments) |
| Something-Something V1[77] | 2017 | 174 | - | - | 108,499 | - | RGB | Human performing actions with everyday objects |
| Kinetics-skeleton[295] | 2017 | 400 | - | - | 260,232 | - | 2DJoints | Human-centered actions |
| HACS[319] | 2017 | 200 | - | - | 1,500,000 | - | RGB+Flow | Human actions (*e.g.*, dancing) |
| Charades-Ego[213] | 2018 | 157 | 112 | 2 | 68,536 | Head-mounted+standard camera | RGB | Egocentric indoor activities |
| AVA[79] | 2018 | 80 | - | - | 211,000 | - | RGB+Flow | Human actions (*e.g.*, talking, sitting) |
| Diving48[133] | 2018 | 48 | - | - | 18,404 | - | RGB+Flow | Diving actions |
| Epic-Kitchens[38] | 2018 | 149 | 32 | - | 39,594 | - | RGB+Flow | Cooking |
| Something-Something V2[77] | 2018 | 174 | - | - | 220,847 | - | RGB | Human performing actions with everyday objects |
| MiT[165] | 2018 | 339 | - | - | 1,000,000+ | - | RGB+Audio+Flow | Dynamic actions (*e.g.*, human, animals) |
| Kinetics-600[19] | 2018 | 600 | - | - | 495,547 | - | RGB | Human-centered actions (*e.g.*, playing instruments) |
| NTU RGB+D 120[142] | 2019 | 120 | 106 | 155 | 114,480 | Kinect v2 | RGB+Depth+3DJoints+Infrared | Daily actions, health-realted actions *etc.* |
| IITR-IAR[92] | 2019 | 21 | 35 | - | 1,470 | FLIR T1020 | Infrared | Human actions (hugging, fighting) |
| Kinetics-700[20] | 2019 | 700 | - | - | 650,317 | - | RGB | Human-centered actions (*e.g.*, playing instruments) |
| HowTo100M[162] | 2019 | 23,611 | - | - | 136,000,000 | - | RGB | Instructional videos (*e.g.*, cooking) |
| CATER[71] | 2019 | 301 | - | - | 5,500 | - | RGB | Compositional actions and temporal reasoning |
| FineGym[206] | 2020 | 530 | - | - | 32,697 | - | RGB | Gymnasium videos (*e.g.*, balance beam) |
| PKU-MMD II[144] | 2020 | 41 | 13 | 3 | 1,009 | Kinect v2 | RGB+Depth+Infrared+3DJoints | Human actions (*e.g.*, standing) |
| EPIC-KITCHENS-100[39] | 2020 | 4,053 | 37 | - | 89,977 | GoPro Hero7 Black | RGB+Flow | Cooking |
| UAV-Human[127] | 2021 | 155 | 119 | - | 22,476 | UAV Camera | RGB+3DJoints | Human Actions (*e.g.*, walking, jogging) |

However, challenges such as noisy labels and class imbalance in real-world datasets can degrade performance, necessitating robust loss functions and data augmentation techniques [36, 75, 307].

Unsupervised learning, by contrast, eliminates the reliance on labels and aims to learn generalizable representations. For example, methods like MoCo [61] and BYOL [78] use contrastive learning to distinguish video instances based on their spatiotemporal features. These methods benefit from diverse datasets with varied contexts, enabling the model to capture a broad range of patterns [69, 195]. However, the lack of labels complicates evaluation, as metrics often depend on downstream tasks [50, 223]. Few-shot and zero-shot learning paradigms address the scarcity of labeled examples [129, 277]. Few-shot methods, such as prototypical networks [217], rely on curated support sets to generalize across classes. Zero-shot approaches [2, 106], powered by vision-language models, use textual descriptions to infer knowledge about unseen actions. For example, CLIP [184] can recognize actions like "playing guitar" by aligning visual features with corresponding textual embeddings,

even when such actions are absent in the training data [306]. Self-supervised learning builds on unlabeled data through pretext tasks, such as temporal order prediction or video masking [59, 230]. These tasks encourage the model to learn useful features without explicit supervision. However, the design of pretext tasks must align with downstream objectives; otherwise, the learned representations may not generalize effectively.

**Architectural innovation.** Video datasets, unlike static image datasets such as ImageNet [195], introduce temporal complexity, requiring specialized architectures. The sequential nature of video data drives innovation in model design to capture both spatial and temporal dependencies. Early attempts [63, 67, 101, 312] to adapt 2D CNNs for video processing fall short, as they are ill-equipped to handle temporal relationships. This limitation leads to the development of 3D CNNs and two-stream networks, such as C3D[231] and I3D[21], which either extend convolutional operations into the temporal dimension to capture motion dynamics or model spatial and temporal information separately. More recently, transformers

[7, 54, 170, 177] have emerged as a powerful alternative. Models like TimeSformer[13] and Video Swin Transformer [150] use attention mechanisms to capture long-range temporal dependencies, making them particularly effective for large-scale and complex datasets. These architectures outperform earlier methods in tasks requiring fine-grained temporal reasoning [88, 196, 251].

Multimodal datasets [137, 205, 214, 285] have further driven the design of architectures that integrate multiple data types. For example, models like CLIP [184] and Flamingo [4] fuse video and textual information, enabling cross-modal reasoning. Similarly, methods tailored for RGB-D data (*e.g.*, combining RGB frames with depth maps) use specialized components to process the complementary modalities effectively. Data augmentation and preprocessing also influence architectural choices. For instance, datasets with high variability in lighting, viewpoint, or action dynamics require architectures with robust components like dropout layers or attention mechanisms [39, 77, 142, 205]. Self-supervised models [5, 274] benefit from contrastive augmentation techniques, where diverse crops or temporal shifts enhance the model's ability to learn invariant spatiotemporal features. Finally, the scale of datasets dictates the complexity of models. Large datasets enable the training of deeper architectures with millions of parameters, while smaller datasets necessitate simpler models or the use of transfer learning [1]. Pre-trained models on large visual datasets (*e.g.*, Kinetics [103] or ImageNet [195]) can be fine-tuned to smaller, domain-specific datasets, demonstrating how data availability shapes model design [42]. Representative models include [59, 120, 196, 230, 251, 273, 274, 306].

The journey of action recognition datasets underscores their central role in shaping the field. From early handcrafted descriptors [284] to cutting-edge transformers [13] and multimodal models [272], the evolution of datasets has driven progress in both learning paradigms and architectures. As datasets become increasingly diverse and complex, they will continue to inspire innovations in action recognition, pushing the boundaries of what machines can learn from video data.

## 3.2 From a Model Perspective

The journey of action recognition models has been shaped by the interplay between data characteristics and the demand for capturing spatiotemporal relationships. Early approaches, intermediate innovations, and the latest breakthroughs all reflect how the challenges and opportunities in data have driven model evolution.

**Early models: handcrafted descriptors and motion-aware designs.** Initial attempts at action recognition rely heavily on handcrafted descriptors tailored to conventional RGB videos [118, 190]. These methods focus on extracting spatiotemporal and motion information. For example, spatiotemporal features like 3D-SIFT [202], extended SURF [282], HOG3D [107], and local trinary patterns [309] are developed to analyze relationships across frames. These descriptors effectively capture the dynamics of simple actions (*e.g.*, walking, waving) in controlled settings. However, they struggle with the complexity of real-world videos, particularly when camera motion introduces noise [324]. To address these challenges, dense trajectories [240] and improved dense trajectories [242] emerge as robust solutions. By tracking local features through video frames, these methods mitigate the impact of camera motion and enabled

better representation of dynamic actions. Bag-of-visual-words [178] and Fisher vector embeddings [119] further enhance their effectiveness, allowing these descriptors to achieve significant success despite limited training data.

**Deep learning revolution: spatiotemporal feature learning.** The advent of large-scale datasets like Sports-1M and Kinetics-400 catalyzes a paradigm shift toward learned feature representations [233]. Inspired by the success of 2D CNNs in image recognition, researchers initially explore 2D networks with temporal aggregation, such as CNN-LSTM[312] and TSN[265], which fuse spatial features across frames. However, these methods lack the capacity to fully capture temporal dynamics [276].

To overcome these limitations, models like two-stream ConvNets [215] and 3D CNNs (*e.g.*, C3D[231] and I3D[21] are introduced. Two-stream architectures use separate branches for spatial and motion information, often using optical flow [116, 261] for the motion stream. Meanwhile, 3D CNNs extend convolutional operations into the temporal dimension, directly modeling spatiotemporal features [183]. Despite their success, both approaches face challenges: two-stream models incur high computational costs [134], while 3D CNNs require extensive data and computational resources [108].

Innovations like (2+1)D convolution decompose 3D operations into separate spatial and temporal components, balancing efficiency and performance [99, 238]. Examples include R(2+1)D networks[234] and their integration with transformers [97], which enhance the ability to model long-range temporal dependencies.

**Transformer era and multimodal integration.** Transformers have redefined action recognition by introducing global attention mechanisms [222]. Vision transformers (ViTs) initially demonstrate the potential for spatial feature extraction in videos [51]. Subsequent transformer-based video models, such as TimeSformer[13] and Motionformer [177], extend this approach to capture complex spatiotemporal relationships. These models excel at handling diverse data distributions and variability in lighting, scale, and viewpoint [88, 129, 251].

Recent advancements include video masked autoencoders (*e.g.*, VideoMAE [230] and VideoMAE V2 [251]), which use self-supervised learning to extract spatial and temporal representations. These architectures, inspired by masked autoencoders in image tasks [84], have set new benchmarks in efficiency and performance for video analysis. Simultaneously, multimodal models such as CLIP [184] and BLIP [124] have integrated video and text data, unlocking new capabilities in action recognition. By aligning video frames with textual descriptions, these models facilitate tasks like zero-shot action recognition and general-purpose video understanding [106, 135, 200]. This integration has paved the way for applications extending beyond action recognition, including video captioning and anomaly detection [45, 47, 296, 325].

**Expanding modalities: depth, skeleton, and large foundation models.** The introduction of depth videos and skeleton sequences through devices like the Microsoft Kinect expands the scope of action recognition [185]. Depth-based models, such as HON4D [174] and HOPC [187], effectively segment human subjects in cluttered scenes, while skeleton-based models capitalize on the structural and temporal continuity of 3D joint movements [53, 300, 322]. Handcrafted skeleton features (*e.g.*, LARP-SO [236])

evolve into learned representations like ST-GCN [295] and its successors [34, 121, 140, 169, 219, 220, 254, 259, 301, 304, 323], including ShiftGCN [30] and CTR-GCN [27]. These graph-based models advance the field by using human pose information for more accurate action recognition. Point cloud-based methods include [10, 164, 275].

Large foundation models like InternVideo2 [273] represent the latest milestone in action recognition. Trained on vast, multimodal datasets, these models demonstrate exceptional versatility across video processing tasks [245, 272, 274]. They exemplify how increased data volume and multimodal integration enable the development of deeper, more powerful architectures, bridging the gap between specialized tasks and general video understanding [5, 46].

**Insights.** The evolution of action recognition models underscores a recurring theme: data characteristics dictate model design. Early handcrafted methods prioritize robustness to motion noise, while deep learning models embrace scale and diversity [175]. Transformers and multimodal architectures have further transformed the field, emphasizing the importance of flexibility and scalability [272, 273]. As video data continues to grow in complexity and volume, future models must navigate challenges such as motion diversity, temporal resolution, and ethical considerations in data use. This journey, driven by both data availability and computational advances, highlights the symbiotic relationship between datasets and model architectures in shaping the trajectory of action recognition.

## 3.3   From a Learning Perspective

The evolution of action recognition models is closely tied to the development of learning paradigms, each offering unique insights and solutions to the challenges posed by video data. From supervised methods relying on large labeled datasets to emerging paradigms like self-supervised and zero-shot learning, the journey reflects a dynamic interplay between data availability, model architecture, and task complexity.

**The supervised learning era.** Supervised learning has been the dominant paradigm in action recognition for decades [67, 101, 231, 326]. Early models rely on fully labeled datasets, where each video is paired with a specific label, such as an action category or bounding box. This explicit mapping between inputs and outputs, guided by loss functions like cross-entropy, enables models to learn spatiotemporal patterns effectively [318]. However, the reliance on high-quality labeled datasets introduces limitations [105]. Labeling video data is costly, time-consuming, and prone to biases, such as noisy labels or skewed class distributions, which degrade model performance. Despite these challenges, supervised learning establishes foundational architectures, including convolutional neural networks (CNNs) [49, 289, 312] and two-stream networks [134, 264, 321], that excel in tasks requiring spatial and motion analysis. Pretraining on large-scale datasets like Kinetics [19, 20, 103] allows models to capture diverse motion patterns, reducing the need for task-specific data through transfer learning [21]. This paradigm demonstrates how large labeled datasets can accelerate progress but also highlights the necessity for alternative approaches to address scalability and diversity challenges.

**The rise of self-supervised and semi-supervised learning.** To overcome the dependence on labeled data, self-supervised learning emerges as a powerful alternative [12, 81]. In this paradigm, models generate pseudo-labels from the data itself, using auxiliary tasks such as predicting motion trajectories [292], solving spatiotemporal puzzles [237], or reconstructing masked regions [230]. Methods like contrastive learning (*e.g.*, SimCLR [26], MoCo [61] and video masked autoencoders (*e.g.*, VideoMAE [230]) demonstrate the ability to learn high-quality spatiotemporal features without explicit supervision [251]. These approaches use data augmentation to create positive and negative pairs, enabling models to distinguish between similar and dissimilar samples [308].

Self-supervised learning has proven particularly effective for pretraining on large-scale unlabeled datasets, significantly enhancing performance on downstream tasks like action recognition. For instance, VideoMAE models, pretrained on small datasets like HMDB51, achieve competitive results, showcasing the paradigm's efficiency in using limited data [230, 251]. Semi-supervised learning bridges the gap between supervised and self-supervised approaches by combining small amounts of labeled data with large volumes of unlabeled data [98, 216, 286, 291]. This paradigm reduces the reliance on extensive labeling efforts, using labeled examples to guide the learning of representations from unlabeled data. Semi-supervised techniques have proven valuable in scenarios where labeled video data is scarce or expensive to obtain.

**Emerging paradigms: few-shot, zero-shot, and unified learning.** Recent advancements [2, 200, 290, 310] have focused on making action recognition models more flexible and adaptable. Few-shot learning enables models to generalize to new action categories using only a handful of labeled examples. Architectures like prototypical networks [217] and relation networks [226] are designed to perform well under limited data conditions, using meta-learning principles. Zero-shot learning goes a step further, enabling models to classify unseen action categories using multimodal inputs, such as textual descriptions or video-text pairs [106]. Models like CLIP [184] demonstrate the effectiveness of vision-language pretraining in achieving generalization across tasks.

Transformers have been instrumental in advancing these paradigms [13, 170]. Originally developed for natural language processing [283], transformers excel in multimodal and unified learning settings. Their attention mechanisms capture long-range dependencies, enabling robust temporal dynamics modeling [35, 172]. By integrating vision and text modalities, transformers facilitate cross-domain learning, paving the way for unified multimodal frameworks capable of handling diverse tasks, from action recognition to video question answering [313].

**Insights.** The trajectory of action recognition learning paradigms underscores the evolving role of data. Labeled datasets have driven supervised learning, while unlabeled and multimodal datasets fuel the rise of self-supervised, semi-supervised, and zero-shot approaches [175]. The interplay between data characteristics and learning methods has shaped models, from CNNs to vision transformers [7, 13, 49]. Future innovations will likely focus on unified learning paradigms that integrate multimodal data and use pretrained video foundation models for broader generalization across tasks.

## 4 Future Directions

In this section, we highlight three key areas poised to shape the future of action recognition: multimodal integration, transformer-based architectures, and vision-language models (VLMs). These directions not only aim to enhance model performance but also tackle some of the most pressing challenges in video understanding.

**Integration of multimodal data.** As video data alone often fails to capture the full complexity of actions, integrating multimodal data (visual, auditory, and textual) has become a critical focus in advancing action recognition. This integration enables models to use complementary information, such as speech, environmental sounds, or contextual text, to better understand actions in diverse and noisy settings. For example, recognizing an action like "talking on the phone" becomes more accurate when the auditory signal (speech) is paired with visual information (body language). The ability to simultaneously process multiple data streams presents new challenges in synchronizing and aligning heterogeneous modalities, but the potential for more robust and nuanced action recognition is vast. This shift to multimodal systems may help models understand actions with greater contextual awareness, reducing ambiguity and improving performance in real-world applications where visual cues alone are often insufficient.

**Transformer-based architectures.** The rise of transformer-based architectures represents a monumental shift in how temporal dependencies are modeled in action recognition. Unlike traditional CNNs, which rely on local spatial filters, transformers excel at capturing long-range dependencies across sequences, making them ideal for video data where context over time is crucial. Transformers enable better modeling of complex temporal dynamics, such as long-range interactions between frames or global motion patterns that span the entire video. By using self-attention mechanisms, transformers can selectively focus on relevant parts of the video sequence, allowing for more accurate action classification, even in the presence of noise or occlusions. This ability to handle long-range dependencies also opens the door to more sophisticated methods for action recognition in dynamic and highly variable environments, such as sports or surveillance footage, where actions are often interdependent and occur over extended periods. While transformer models are computationally intensive, their increasing efficiency and scalability make them a promising avenue for the next generation of action recognition systems.

**Vision-language models.** Another transformative trend in action recognition is the integration of vision-language models (VLMs), which combine the understanding of visual content with linguistic representations. These models have the potential to overcome one of the biggest challenges in action recognition: understanding ambiguous or context-dependent actions. By incorporating natural language processing (NLP) techniques, VLMs can infer the meaning behind a sequence of actions in a video based on textual descriptions or situational context. For instance, the action of "grabbing a cup" could be interpreted differently based on the surrounding environment or verbal cues, such as "grabbing a cup of coffee" versus "grabbing a cup to throw". This alignment between vision and language facilitates more comprehensive reasoning about actions and allows models to handle complex, abstract tasks like action sequencing, goal recognition, and activity prediction. Furthermore, VLMs enable the development of systems that can interact with users or adapt to specific contexts, making them highly applicable for interactive media, autonomous systems, and personalized healthcare applications.

**Potential for cross-domain advancements.** The integration of these emerging trends also opens new opportunities for cross-domain advancements in action recognition. Multimodal data and transformer architectures, for instance, can be combined to tackle complex video datasets where both long-range temporal dependencies and multimodal context are essential. Similarly, VLMs can be enhanced with transformer-based architectures to refine the attention mechanisms, improving both the understanding of temporal dynamics and the contextual alignment between visual and linguistic data. These hybrid approaches not only promise to address current challenges but also pave the way for a new generation of action recognition systems that are adaptable, context-aware, and capable of reasoning about actions in a human-like manner.

The future of action recognition lies in the intersection of multimodal data integration, transformer-based architectures, and VLMs. By addressing the challenges of temporal complexity, contextual ambiguity, and cross-domain generalization, these trends have the potential to revolutionize the field, making action recognition more accurate, adaptable, and robust across diverse real-world applications. As these technologies mature, we anticipate a significant leap forward in how video content is understood and processed, leading to more intelligent systems that can interpret, predict, and interact with the world in ways previously imagined only in science fiction.

## 5 Conclusion

Action recognition has evolved significantly, driven by advancements in data, model architectures, and learning paradigms. Initially relying on handcrafted features and small labeled datasets, the field shifted with the advent of large-scale video datasets and learned representations, using models like 2D, 3D, and (2+1)D CNNs, and GCNs. As video data grow more complex, innovative learning paradigms, such as self-supervised, few-shot, and contrastive learning, help harness the power of large, unlabeled datasets. The introduction of transformer-based models marks a key milestone, enhancing the ability to capture temporal dynamics. Masked autoencoders improve the balance between spatial and temporal features, while the integration of language models enriched action recognition with semantic context. The rise of video foundation models, combining image, video, and language data, has expanded the scope of action recognition to include broader video processing tasks, such as anomaly detection and video captioning. Ultimately, the evolution of action recognition has transformed it into a core element of general video processing, offering insights for future challenges and opportunities in video analysis and beyond.

## Acknowledgments

Xi Ding, a Research Assistant with the Temporal Intelligence and Motion Extraction (TIME) Lab at ANU, contributed to this work. This research was conducted under the supervision of Lei Wang.

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
