# OpenReview forum: "The Journey of Action Recognition"
_ACM.org/TheWebConf/2025/Workshop/TIME — TIME 2025 Oral_

### Official Review · Reviewer_preJ · 2025-01-11
**The review evaluates a comprehensive work on action recognition, highlighting its quality, clarity, originality, and significance. The paper is praised for its holistic approach, integrating data, learning paradigms, and model architectures, and offering a forward-looking perspective on the field. It emphasizes the transition from handcrafted features to learned representations and explores the potential of vision-language models and transformers. While the work is significant and insightful, it lacks specific experimental results, which are crucial for assessing the impact of proposed methodologies. Overall, it is a valuable contribution, providing a roadmap for future developments in action recognition.**

**Rating:** 8
**Confidence:** 3

**Review:**

Quality: The work appears to be of high quality, as it integrates various aspects of action recognition into a unified framework. It provides a comprehensive analysis of the interplay between data, learning paradigms, and model architectures which is a significant contribution to the field.

Clarity: The paper is clear in its objectives and scope. It distinguishes itself from existing surveys by offering a holistic approach and forward-looking perspective, particularly in the integration of vision language models and transformers.

Originality: The originality of the work lies in its holistic approach to action recognition, integrating data, learning paradigms and model architectures. It also explores underexplored areas such as the potential of vision-language models and transformers for cross-modal learning.

Significance: The work is significant as it not only reflects on past advancements but also provides a roadmap for future developments in action recognition. It highlights the transformative role of large language models and the integration of multimodal, temporal, and semantic information.

Overall Pros:

Comprehensive analysis of the evolution of action recognition. Integration of data, learning paradigms, and model architectures. Forward-looking perspective on the role of vision-language models and transformers. Highlights the transition from handcrafted features to learned representations. Discusses the potential of self-supervised and semi-supervised learning.

Overall Cons:

The context does not provide specific experimental results or quantitative evaluations which are typically important for assessing the impact of proposed methodologies. The paper may be dense for readers not familiar with the technical aspects of action recognition, as it covers a wide range of topics. Overall, the work is a valuable contribution to the field of action recognition, offering insights into both historical and future trends.

---

### Official Review · Reviewer_PKhU · 2025-01-13

**Rating:** 8
**Confidence:** 3

**Review:**

Auther has include some original experimental results, create a new taxonomy, or propose a unique framework for action recognition to add novelty.
Also they Took a closer look at the limitations of existing methods and datasets, and suggest practical ways to tackle issues like scalability or the high costs of annotation.
Discussed privacy concerns, potential biases, and the broader societal impact of action recognition systems in more depth.

---

### Official Review · Reviewer_SuRR · 2025-01-19
**Review of The Journey of Action Recognition**

**Rating:** 8
**Confidence:** 3

**Review:**

This paper provides a comprehensive summary of the development in action recognition, analyzing it from the perspectives of data, model architecture, and learning paradigms. The author begins by introducing early manual feature modeling methods and then offers a detailed overview from the three dimensions of data, model architecture, and learning paradigms. For example, in the model architecture section, the paper starts with early 2D networks, then moves on to Transformer-based models for capturing long-term temporal dependencies, and finally discusses how VLMs (Vision-Language Models) have further advanced the field, providing valuable insights for the development of action recognition. The advantages of the paper are as follows:
- Rich content: The author analyzes the development of action recognition from multiple perspectives, providing significant value for the future development of the field.
- In-depth analysis: The author deeply analyzes the field of action recognition from the angles of data, model architecture, and learning paradigms, highlighting key milestones and breakthroughs, and offering forward-looking insights on the role of large models in advancing the field.

However, the only drawback of the paper is the lack of an open-source repository to organize and summarize the relevant literature, as well as the absence of a comparative experimental summary of results from different models.

---

### Meta-Review · Area_Chair_ZcMY · 2025-01-26

**Recommendation:** Accept (Oral)
**Confidence:** 5

**Metareview:**

The paper presents a methodologically rigorous survey of the evolution of action recognition, model architectures, and learning paradigms. It outlines how the field has progressed from handcrafted features and simple datasets to sophisticated deep-learning models like 3D CNNs. The integration of multimodal data and novel learning approaches, such as self-supervised and zero-shot learning, demonstrates the authors’ deep understanding of the field.

As a survey paper, the author provided a cohesive overview of advancements and highlighted trends, challenges, and future directions. The progression of ideas is logically structured, providing a clear timeline of advancements in action recognition. The author provided learning paradigms with detailed comparisons between supervised, self-supervised, and zero-shot approaches.

Overall Cons:
1. Limited critical analysis of current methods’ real-world challenges, such as computational scalability and ethical considerations.
2. The benchmarking discussion could expand on dataset biases, such as the over-reliance on benchmarks like Kinetics and UCF101.

Overall Evaluation:
The paper provides a comprehensive synthesis of the field, and the evolution of action recognition, making it a valuable resource for both newcomers and experienced researchers. Its focus on deep learning models and novel learning paradigms is well-aligned with advancements in AI. While the discussion on benchmarks and challenges could be expanded, the paper provides enough depth and breadth to justify the presentation.

Recommend it for an oral presentation.

---

### Decision · Program_Chairs · 2025-01-27

**Decision:**

Accept (Oral)

**Comment:**

This paper consistently received three clear acceptance scores. All reviewers and the area chair appreciated the significance of this work.

The program chair concurs with the area chair's decision.

For the camera-ready version, please revise your paper according to the feedback provided by the reviewers.

Workshop papers must be written in English, follow a double-column format, and comply with the [ACM template](https://www2025.thewebconf.org/short-papers) and formatting guidelines. The template is also available in [Overleaf](https://www.overleaf.com/latex/templates/association-for-computing-machinery-acm-sig-proceedings-template/bmvfhcdnxfty). For authors using Microsoft Word, the Word Interim Template is recommended.

Camera-ready versions of accepted papers can and should include all information to identify authors, and should acknowledge any funding received that directly supported the presented research.

In addition, ensure that the DOI (to be provided by the PCs at a later stage) is included, and cite the workshop (to appear) using the following reference:

```
@inproceedings{time2025,
  title={TIME 2025: 1st International Workshop on Transformative Insights in Multi-faceted Evaluation},
  author={Lei Wang and Md Zakir Hossain and Syed Islam and Tom Gedeon and Sharifa Alghowinem and Isabella Yu and Serena Bono and Xuanying Zhu and Gennie Nguyen and Nur Haldar and Seyed Jalali and Abdur Razzaque and Imran Razzak and Rafiqul Islam and Shahadat Uddin and Naeem Janjua and Aneesh Krishna and Manzur Ashraf},
  booktitle={ACM Web Conference Workshop},
  year={2025}
}
```

Please note that at least one in-person registration is required for each accepted workshop paper to be included in the Companion Proceedings of WWW 2025. All accepted papers must be presented at the conference. Papers not presented (no-shows) may be withdrawn from the companion proceedings. Presentations will be conducted in two formats: oral and poster.

The camera-ready deadline for workshop papers is 7 February 2025 (AoE).